# Mitochondrial Fraction of Circulating Cell-Free DNA as an Indicator of Human Pathology

**DOI:** 10.3390/ijms25084199

**Published:** 2024-04-10

**Authors:** Maria Panagopoulou, Makrina Karaglani, Konstantina Tzitzikou, Nikoleta Kessari, Konstantinos Arvanitidis, Kyriakos Amarantidis, George I. Drosos, Spyros Gerou, Nikolaos Papanas, Dimitrios Papazoglou, Stavroula Baritaki, Theodoros C. Constantinidis, Ekaterini Chatzaki

**Affiliations:** 1Laboratory of Pharmacology, Department of Medicine, Democritus University of Thrace, 68100 Alexandroupolis, Greecetzitzikouu@gmail.com (K.T.);; 2Institute of Agri-Food and Life Sciences, University Research and Innovation Centre, Hellenic Mediterranean University, 71003 Heraklion, Greece; 3Clinic of Medical Oncology, Department of Medicine, Democritus University of Thrace, University General Hospital of Alexandroupolis, 68100 Alexandroupolis, Greece; 4Clinic of Orthopaedic Surgery, Department of Medicine, Democritus University of Thrace, University General Hospital of Alexandroupolis, 68100 Alexandroupolis, Greece; 5Analysis Biopathological Diagnostic Research Laboratories, 54623 Thessaloniki, Greece; 6Diabetes Centre, 2nd Department of Internal Medicine, University Hospital of Alexandroupolis, 68100 Alexandroupolis, Greece; 7Laboratory of Experimental Oncology, Division of Surgery, School of Medicine, University of Crete, 71500 Heraklion, Greece; 8Laboratory of Hygiene and Environmental Protection, Department of Medicine, Democritus University of Thrace, 68100 Alexandroupolis, Greece

**Keywords:** mitochondrial, ccfDNA, prediabetes, breast cancer, diabetes, osteoarthritis, machine learning

## Abstract

Circulating cell-free DNA (ccfDNA) of mitochondrial origin (ccf-mtDNA) consists of a minor fraction of total ccfDNA in blood or in other biological fluids. Aberrant levels of ccf-mtDNA have been observed in many pathologies. Here, we introduce a simple and effective standardized Taqman probe-based dual-qPCR assay for the simultaneous detection and relative quantification of nuclear and mitochondrial fragments of ccfDNA. Three pathologies of major burden, one malignancy (Breast Cancer, BrCa), one inflammatory (Osteoarthritis, OA) and one metabolic (Type 2 Diabetes, T2D), were studied. Higher levels of ccf-mtDNA were detected both in BrCa and T2D in relation to health, but not in OA. In BrCa, hormonal receptor status was associated with ccf-mtDNA levels. Machine learning analysis of ccf-mtDNA datasets was used to build biosignatures of clinical relevance. (A) a three-feature biosignature discriminating between health and BrCa (AUC: 0.887) and a five-feature biosignature for predicting the overall survival of BrCa patients (Concordance Index: 0.756). (B) a five-feature biosignature stratifying among T2D, prediabetes and health (AUC: 0.772); a five-feature biosignature discriminating between T2D and health (AUC: 0.797); and a four-feature biosignature identifying prediabetes from health (AUC: 0.795). (C) a biosignature including total plasma ccfDNA with very high performance in discriminating OA from health (AUC: 0.934). Aberrant ccf-mtDNA levels could have diagnostic/prognostic potential in BrCa and Diabetes, while the developed multiparameter biosignatures can add value to their clinical management.

## 1. Introduction

Circulating-cell-free DNA (ccfDNA) is DNA liberated from cells into biological fluids, e.g., blood, lymph, bile and urine [1]. It is double- or single-stranded and can be released during apoptosis, necrosis, or even by active mechanisms. In health, ccfDNA mainly derives from tissues and cells like hematopoietic, whereas in some diseases, it is highly enriched from pathological tissues. In cancer, ccfDNA is liberated from tumor cells, metastatic sites and CTCs. In general, ccfDNA has been proven to reflect dynamically the genetic and epigenetic events taking place in a tissue or tumor [2,3].

ccfDNA is mainly considered to be of nuclear origin. However, mitochondria can also contribute their own circular double-stranded genome to the circulation. Mitochondrial DNA (mtDNA) contains 13 protein-encoding genes that are essential for the electron transport chain and ATP synthase [4]. It has been reported that ccfDNA of mitochondrial origin (ccf-mtDNA) consists of shorter DNA fragments different than those of the nuclear ccfDNA (ccf-nDNA) [5,6]. Furthermore, ccf-mtDNA may be found in low abundance due to its higher susceptibility to degradation lacking histone protection [7]. Still, evidence suggests that the quantity of ccf-mtDNA or its other features could have a diagnostic or prognostic potential in several diseases. For example, levels of ccf-mtDNA were quantified to be elevated in different cancer types [8,9,10,11], including Breast Cancer (BrCa) [12,13], where higher levels were correlated to unfavorable clinicopathological characteristics [13,14]. Aberrant ccf-mtDNA levels have also been reported in other diseases, such as Alzheimer’s [15]. In type 2 Diabetes (T2D), ccf-mtDNA levels were elevated as compared to health [15,16], being even higher for those patients also suffering from Coronary Heart Disease. In addition, modified levels of ccf-mtDNA were observed in patients with Cardiovascular diseases [17].

Here, we introduce a simple and effective standardized Taqman probe-based dual-qPCR assay for the simultaneous detection and relative quantification of nuclear and mitochondrial fragments of ccfDNA. Three pathological entities of major burden, i.e., one malignancy (BrCa), one inflammatory (osteoarthritis, OA) and one metabolic (T2D), were selected as use cases to validate the diagnostic application of ccf-mtDNA quantity in human pathology. Upon assay measurements of patient blood samples, automated Machine Learning was then used to analyze our experimental data in order to build diagnostic biosignatures of increased performance. The experimental workflow is depicted in Figure 1.

## 2. Results

### 2.1. Development of Dual PCR Assay for Detecting ccf-mtDNA and Reference Measurements

The sensitivity of the dual PCR assay was assessed by serial dilutions of a commercial genomic DNA mix containing both nuclear and mitochondrial DNA (100 ng, 10 ng, 1 ng, 0.1 ng, 0.01 ng). Both nuclear *GAPDH* and mitochondrial *MTATP8* gene targets were detected in down to 0.01ng of genomic DNA. The efficiency of *GAPDH* and *MTATP8* was 96% and 100%, respectively. In Figure 2, the standard and amplification curves of the *GAPDH* and *MTATP8* assays are presented. Results were expressed as RQ_sample_ (Relative Quantification) of the mtDNA fraction in relation to nuclear. There was no correlation to sex or age in the reference values measured in the healthy individuals’ group. 

### 2.2. ccf-mtDNA Fraction in Diseases

#### 2.2.1. BrCa

Total plasma ccfDNA levels and their relative content in ccf-mtDNA were measured in 119 BrCa patients and in 77 healthy female individuals and are presented in Table 1. Confirming previous results [18], levels of total ccfDNA were significantly elevated in all BrCa patients as compared to health (*p* = 0.005, Mann–Whitney U = 564). Interestingly, those increased ccfDNA levels in BrCa were also more abundant in their ccf-mtDNA content, as the later was found to be higher in the BrCa patients as compared to healthy individuals (*p* < 0.001, Mann–Whitney U = 2181) (Table 1 and Figure 3A). In particular, in the Adjuvant and Metastatic BrCa groups, ccf-mtDNA content was statistically significantly higher (*p* < 0.001, Mann–Whitney U = 1282 and *p* = 0.044, Mann–Whitney U = 429, respectively) (Figure 3B) as compared to health, with the group of the Adjuvant presenting the highest levels among the three groups. A positive correlation was observed between the ccf-mtDNA fraction and plasma total ccfDNA in the BrCa patients (*p* = 0.047, r = 0.186), but not in the healthy individuals. Next, ccf-mtDNA content was correlated to the patients’ clinicopathological features. In the Adjuvant group, the absence of HER-2/neu expression was correlated with a higher ccf-mtDNA fraction (*p* = 0.05, Mann–Whitney U = 364) (Figure 3C). Similarly, in the Metastatic group, elevated ccf-mtDNA content was correlated to the absence of HER-2/neu expression (*p* = 0.006, Mann–Whitney U = 5) (Figure 3D) and to the presence of ER receptor (*p* = 0.028, Mann–Whitney U = 7) (Figure 3E). No other correlations were observed between ccf-mtDNA quantity and age, menopause, receptor status, grade, stage or nodal status.

#### 2.2.2. Diabetes and Prediabetes

The quantity of total plasma ccfDNA and its relative content in ccf-mtDNA were measured in 158 T2D patients, in 78 Prediabetic individuals and in 100 Healthy individuals, as shown in Table 2. Levels of total ccfDNA were significantly higher in both patient groups as compared to the healthy individuals (*p* < 0.001, Kruskal–Wallis df = 2), and they did not significantly differ between diabetes and prediabetes. Following the same pattern, ccf-mtDNA content was also higher in Diabetes and Prediabetes in relation to Health (*p* = 0.008, Mann–Whitney U = 1591 and *p* < 0.001, Mann–Whitney U = 821, respectively) (Figure 4) and did not differ between the two patient groups. A weak positive correlation was observed between the fraction of ccf-mtDNA and total plasma ccf-DNA in the Diabetes group (*p* = 0.04, r = 0.252). Also, an inverse correlation was observed between age and ccf-mtDNA content, but only in the Prediabetes group (*p* < 0.001, r = −0.385). No other significant correlations were observed between ccf-mtDNA content and sex, HbA1c, glucose or BMI.

#### 2.2.3. Osteoarthritis

The quantity of total plasma ccfDNA and its relative content in ccf-mtDNA were measured in 17 female OA patients and in 17 Healthy women, and the results are presented in Table 3. Levels of total ccfDNA were significantly higher in OA as compared to the Healthy group (*p* < 0.001, *t*-test df = 29). Ccf-mtDNA content did not differ between OA and Health (*p* = 0.852, *t*-test df = 29) (Figure 5). Similarly, no other significant correlation was observed between ccf-mtDNA content and age or BMI.

### 2.3. AutoML Multivariate Analysis

#### 2.3.1. BrCa

Demographical and experimental data from 119 BrCa patients and 77 Healthy individuals were uploaded to JADBio in order to build a biosignature for discriminating between Adjuvant, Neoadjuvant, Metastatic and Healthy groups. AutoML classification analysis resulted in a biosignature of five features, including plasma ccf-DNA, extracted ccf-DNA, age and RQ of ccf-mtDNA via a support vector machines (SVM) algorithm (https://app.jadbio.com/share/1e48c72b-dd55-4151-a1e6-6382f678c9c0, accessed on 9 April 2023). In discriminating groups, this signature reached an area under the curve (AUC) of 0.698 (0.634, 0.732) and an average precision of 0.789 (0.754, 0.819) (Figure 6A,B). Next, another classification analysis was performed in order to build a biosignature for classifying between BrCa and Health. A three-feature biosignature emerged, including plasma ccf-DNA, extracted ccf-DNA and RQ of ccf-mtDNA via a SVM algorithm (https://app.jadbio.com/share/04ffa028-dbde-4476-b1bc-8a55348e8c60, accessed on 9 April 2023). The signature showed an AUC of 0.887 (0.841, 0.930) and an average precision of 0.902 (0.864, 0.938) (Figure 6C,D). From the Feature Importance plots, it becomes apparent that the total amount of ccfDNA is the decisive discriminative feature, with the rest contributing minimally to the models’ performance. Finally, survival analysis was conducted in the Adjuvant group. A biosignature of five features emerged, containing the Disease Free Interval (DFI), grade, extracted ccf-DNA, age and RQ of ccf-mtDNA via a Survival Random Forest (SRF) algorithm (https://app.jadbio.com/share/ce99f9ad-c4dc-426f-b4ca-92b1ae431840, accessed on 9 April 2023) for predicting overall survival in the Adjuvant group of patients. The Concordance Index of the signature was 0.756 (0.691, 0.819) (Figure 6E,F).

#### 2.3.2. Diabetes and Prediabetes

Demographical and experimental data from 178 Diabetic patients, 58 Prediabetic individuals and 100 Healthy individuals were uploaded to JADBio in order to build a biosignature for discriminating among the three groups. AutoML classification analysis resulted in a biosignature of five features, including plasma ccf-DNA, extracted ccf-DNA, sex, age and RQ of ccf-mtDNA via a Classification Random Forest (CRF) algorithm (https://app.jadbio.com/share/f735cc79-d31d-4e3e-abf7-7aceb9d3bb1d, accessed on 9 April 2023). Signature’s performance showed an AUC of 0.772 (0.726, 0.815) and an average precision of 0.808 (0.770, 0.844) (Figure 7A,B). Furthermore, a signature for discriminating between Diabetic patients and Healthy individuals was constructed containing the same five features as above via a CRF algorithm (https://app.jadbio.com/share/32882c69-b4bb-4d7c-bf6c-86822e3a7cab, accessed on 9 April 2023). The AUC of biosignature was 0.797 (0.745, 0.843), and the average precision was 0.820 (0.781, 0.857) (Figure 7C,D). Next, a four-feature signature for identifying Prediabetes from Health was built, including plasma ccf-DNA, extracted ccf-DNA, age and RQ of ccf-mtDNA, again via the CRF algorithm (https://app.jadbio.com/share/31fe2b21-3fa9-4a46-ade5-61f8aa1ea2d7, accessed on 9 April 2023). Signature’s performance reached an AUC of 0.795 (0.743, 0.848) and average precision of 0.836 (0.798, 0.875) (Figure 7E,F). Interestingly, in comparison to the BrCa results, ccf-mtDNA revealed a higher contribution to the models’ performance along with the total ccfDNA quantity, especially in the case of prediabetes.

#### 2.3.3. Osteoarthritis

Regarding OA, the demographic and experimental data of 17 female patients and 17 healthy women were uploaded to JADBio, and this classification analysis resulted in a one-feature biosignature including plasma ccfDNA via the SVM algorithm (https://app.jadbio.com/share/9db37c36-8f6e-4aff-9685-4cc71cf6e6cb, accessed on 9 April 2023). Signature reached an AUC of 0.934 (0.771, 1.000) and an average precision of 0.963 (0.877, 1.000) in discriminating OA. 

## 3. Discussion

CcfDNA has been proven to be a valuable biomaterial for clinical applications. Similarly to cellular DNA, mitochondrial DNA represents only a small fraction of total ccfDNA, still attracting interest due to its potential as a clinical circulating biomarker [16,19]. In the present study, we compared the mitochondrial ccfDNA fraction in three different pathological entities: one malignancy (BrCa), one metabolic (T2D) and one inflammatory (OA), in relation to health and other clinical parameters. A dual PCR-based assay was developed, targeting both nuclear and mitochondrial genes in the same reaction. Following this, AutoML technology was applied to our experimental data in combination with patients’ demographic and clinical data to build specific diagnostic and prognostic biosignatures for each medical condition and highlight their potential clinical significance. 

In BrCa, the fraction of ccf-mtDNA was found to be higher in relation to health. Pasha et al., had shown that ccf-mtDNA was elevated in BrCa patients in relation to healthy individuals [20]. Another study proved that ccf-mtDNA was elevated in BrCa and that ccf-mtDNA levels were correlated with unfavorable clinical parameters such as grade, stage, lymph node and hormonal receptor status [12]. In accordance with that, data from investigating the mtDNA fraction in whole blood showed that early-stage BrCa patients had a lower content of mtDNA than patients with more advanced disease [14]. Here, we confirm these results, in particular in adjuvant and metastatic patients, also demonstrating correlations to the absence of HER-2/neu and to the presence of ER, findings worthy of further clinical evaluation to demonstrate value as a biomarker in BrCa. In addition, total ccfDNA levels were found to be elevated in our BrCa patient cohort, confirming previous results [18,21]. Interestingly, the higher the levels of total ccfDNA in BrCa, the more abundant its mtDNA fraction. One can speculate that the pathological mechanism resulting in the elevated release of tumor DNA in the circulation favors the mtDNA component. 

Additionally, in our study, the absence of HER-2/neu expression in BrCa patients was correlated with higher ccf-mtDNA quantity in the Adjuvant and Metastatic groups, and the presence of the ER receptor was associated with higher ccf-mtDNA levels in the Metastatic group. This is in accordance with previous studies in BrCa, showing that the status of hormonal receptors is associated with ccf-mtDNA levels. Safi et al., showed a negative correlation between the ccf-mtDNA ratio and PR receptor status [19]. Similarly, Mahmoud et al., found that aberrant levels of ccf-mtDNA were correlated to hormonal receptor status in BrCa patients [12]. Functional studies could shed light on the mechanism by which mtDNA affects hormonal receptors and vice versa. Moreover, it would be interesting to explore the respective correlations in other hormone-dependent malignancies. 

In the Diabetes cohort, we also found that the ccf-mtDNA fraction was higher in T2D and Prediabetes in relation to healthy individuals. In agreement with our findings, Liu et al., found that ccf-mtDNA levels were elevated in T2D. In parallel, the consistent increase of ccf-mtDNA in T2D was associated with the presence of coronary heart disease [20] as well as Alzheimer’s [15]. Yuzefovych et al., showed that ccf-mtDNA fragments were elevated in obese T2D patients in relation to healthy individuals, and there was also a positive correlation between insulin resistance and the abundance of ccf-mtDNA fragments [22]. Rosa et al., found that increased ccf-mtDNA levels in diabetes were correlated with elevated cellular mtDNA levels [23]. Another study suggested that ccf-mtDNA could be evaluated as a biomarker of heart failure in patients with T2D [24]. Given these promising results, ccf-mtDNA could have clinical value in the management of diabetes. Our results reinforce this view, also adding the observation in the prediabetic group, supporting that the molecular events leading to the increased mtDNA fraction in the circulation are related to the initiation of the pathology. 

Although ccfDNA has been reported in the synovial fluid of OA patients [25], according to our knowledge, its presence in the blood has not been previously studied. Here, we show that the levels of total ccfDNA were higher in patients as compared to Healthy controls. Regarding its mitochondrial fraction, we found similar ccf-mtDNA levels between patients and healthy individuals in our small pilot cohort of female patients. We suggest further study of ccf-mtDNA levels in OA enrolling more patients, as our small number of samples is an obvious limitation that may restrict statistical power. Levels of ccf-mtDNA had been previously shown to be elevated in patients with Rheumatoid Arthritis [26,27]. Corroborating our findings, an interesting study in Posttraumatic Osteoarthritis showed elevated levels of ccf-mtDNA in synovial fluid after injury, originating, according to the authors, from cell death but also from active release by chondrocytes [28]. Moreover, they proved that joints with naturally occurring intra-articular fractures presented increased synovial fluid ccf-mtDNA levels and suggested its exploitation as a biomarker of cartilage degeneration [28]. We show here that these differences have also been reflected in the blood mtDNA fraction. 

According to the bibliography, the ccf-mtDNA fraction has been shown to be elevated in multiple pathologies. For instance, levels of ccf-mtDNA are increased in cardiovascular diseases and are found to be correlated to hypercholesterolemia and arterial hypertension [29]. Furthermore, in urological malignancies such as bladder cancer, renal cell carcinoma, and prostate cancer, ccf-mtDNA was also elevated in relation to healthy individuals [9]. Besides plasma or serum, ccf-mtDNA can also be detected in Cerebrospinal Fluid (CSF), and ccf-mtDNA is decreased in the CSF of patients with neurological disorders such as Alzheimer, Parkinson and Multiple Sclerosis [30]. Overall, these and our data show that ccf-mtDNA presents aberrant levels in several pathologies and could serve as a potential biomarker alone or in combination with other markers in the clinical setting.

To further exploit these insights, we conducted a multiparameter analysis incorporating not only experimental measurements but also clinical and demographic data using automated machine learning. In BrCa, two biosignatures were produced, presenting efficiency in discriminating between health and BrCa and predicting the overall survival of patients undergoing adjuvant therapy. The ccf-mtDNA fraction had a minimal contribution to their performance, with the levels of total ccfDNA being the decisive feature, as also shown previously [18]. On the other hand, in the diabetes cohort, three biosignatures were built showing high performance to stratify among Diabetic, Prediabetic and healthy subjects. Here, the ccf-mtDNA content had a stronger contribution to the performance of the model, especially in the case of prediabetes. Finally, although ccf-mtDNA was not selected as a feature, using ad hoc autoML in our small dataset, we were able to build a biosignature of impressive performance in discriminating OA from health based on ccfDNA levels, and this could have translational clinical value. 

## 4. Materials and Methods

### 4.1. Clinical Samples

The study was approved by the Scientific Board of the University General Hospital of Alexnadroupoli (PGNA), following an assessment by the Ethics Committee, and was conducted according to the ethical principles of the 1964 Declaration of Helsinki and its later amendments. All patients participated after signing a voluntary informed consent. In the present study, we enrolled BrCa patients, T2D patients, pre-Diabetes individuals, OA patients and healthy individuals.

#### 4.1.1. BrCa

Plasma samples were collected from 119 BrCa women who visited the Department of Medical Oncology of PGNA and were allocated to three groups: (a) 81 patients having recently (within the previous month) undergone surgery for primary BrCa, exactly before the initiation of adjuvant therapy (Adjuvant group); (b) 17 patients upon diagnosis for BrCa, having no previous surgery, before the initiation of neo-adjuvant therapy (Neo-adjuvant group); (c) 21 patients upon diagnosis for metastatic disease before the initiation of first-line chemotherapy (a combination of Taxane/Anthracyclines) (Metastatic group). The BrCa type was invasive ductal carcinoma for all patients enrolled in the study. The available clinicopathological features are presented in Table 4.

#### 4.1.2. Diabetes and Pre-Diabetes

Plasma samples were obtained from 158 T2D patients and 78 individuals with impaired glucose tolerance (pre-Diabetes) diagnosed according to 2006 WHO recommendations for the diagnostic criteria for diabetes and intermediate hyperglycemia. T2D patients and pre-Diabetes individuals were recruited in the Department of Pathology at PGNA. Available clinicopathological and demographical data are presented in Table 5.

#### 4.1.3. Osteoarthritis

Plasma samples were collected from 17 female patients with OA who had undergone total knee arthroplasty at the Department of Orthopaedics at PGNA. Available clinicopathological and demographical data are shown in Table 6.

#### 4.1.4. Healthy Individuals

Age- and sex-matched healthy individuals were recruited from the Blood Donation Unit of PGNA. In total, plasma samples from 80 women and 82 men were collected. Available demographic characteristics of healthy women used in BrCa analysis are shown in Table 4, healthy men and women included in the Diabetes comparative analysis are shown in Table 5, and demographic characteristics of healthy women used in OA analysis are included in Table 6.

### 4.2. Human Samples Pre-Analytical Procedures

Plasma was isolated within 2 h from blood sampling in EDTA-coated tubes through centrifugation at 2000× *g* for 10 min. An additional high-speed centrifugation step at 14,000× *g* for 10 min was performed to remove any cellular debris and contaminants. Plasma samples were stored at −80 °C until further analysis. Following, direct quantification of total unbounded/naked ccfDNA in 20 μL of plasma was performed utilizing the Quant-iT dsDNA High-Sensitivity Assay kit in the Qubit v3.0 Fluorometer (Invitrogen, Darmstadt, Germany), according to manufacturer specifications. A standard curve was generated using the provided standards (0 and 10 ng/μL). Then, ccfDNA was extracted automatically from 1200 μL of plasma using the MagCore Plasma DNA Extraction kit in the MagCore system (RBRCA Bioscience, New Taipei City, Taiwan) according to the manufacturer’s instructions, and its quantity was estimated by Qubit. Extracted ccfDNA samples were stored at −20 °C until further processing. 

### 4.3. Quantification of ccf-mtDNA

Quantities of ccf-mtDNA and ccf-nDNA fragments were measured by a Taqman probe-based dual-qPCR assay using the nuclear Glyceraldehyd-3-phosphat-dehydrogenase (*GAPDH*) and the mtDNA-encoded ATPase 8 (*MTATP8*) reference genes. Primers for mtDNA and nDNA reference genes are presented in Table 7. Each qPCR was carried out in 20 μL of total reaction volume containing 11.3 μL H_2_O, 4 μL 5X Platinum II buffer, 0.8 μL Mg, 0.5 μL dNTPs mix, 0.4 μL of each GAPDH and MTATP8 primer mix (10 μM), 0.2 μL of a 10 μM FAM-labeled MTATP 8-probe and 0.2 μL of a 10 μM CY5-labeled GAPDH-probe. For each reaction, 2 μL of ccfDNA were added. All qPCR reactions were performed using the Rotor-Gene 6000 Series (Qiagen, Darmstadt, Germany) under the following conditions: an initiation step for 3 min at 95 °C, followed by a first denaturation for 15 s at 95 °C, and a further step consisting of 40 cycles of 50 s at 58 °C. The assay efficiency (expressed as E = 10^−1/slope−1^) was evaluated by using serial dilutions of human genomic DNA (Promega, Madison, WI, USA) in dH_2_O (100–0.01 ng). This DNA consists of both ncDNA and mtDNA in order to resemble the composition of ccfDNA samples that also contain ncDNA and mtDNA fragments. The results were calculated using Rotor-Gene Software 1.7 (Qiagen). The analysis was performed according to the RQ_sample_ (Relative Quantification) = 2^ΔCT^ method. Specifically, ΔCT values were generated using the type ΔCT= Ct_GAPDH_ − Ct_MTATP8_ for each sample.

### 4.4. Statistical Analysis

The Kolmogorov–Smirnov test was used to check for normality in the distribution. A one-way ANOVA test that was followed by a Bonferroni post hoc or Kruskal–Wallis test was applied to compare continuous variables between subgroups. In the case of binary variables, the *t*-test or Mann–Whitney test were also applied. Pearson (R) or Spearman (r) correlation was used for comparison between two continuous variables. All statistical tests employed in our analysis were two-sided. Statistical significance was placed at a *p*-value < 5 × 10^−2^. Continuous variables are expressed as median (minimum-maximum) or mean ± standard deviation. Categorical variables are shown as absolute frequencies. Statistical analysis was conducted with the IBM SPSS 19.0 statistical software (IBM Corp. 2010. IBM SPSS Statistics for Windows, Version 19.0., Armonk, NY, USA).

### 4.5. AutoML Multivariate Analysis

For Machine Learning analyses, we employed the autoML technology of JADBio [31]. JADBio applies to low- or high-sample data, as well as to high-dimensional or low-scale omics data, and produces accurate predictive models estimating the out-of-sample model’s performance after bootstrap correction and cross-validation. Given a 2D matrix, JADBio preprocesses data, including mean imputation, mode imputation, constant removal, and standardization, and then tries several predictive algorithms such as Classification Random Forests, Support Vector Machines, Ridge Logistic Regression, and Classification Decision Trees. Specifically, for small sample sizes, it employs a stratified, K-fold, repeated cross-validation BBC-CV algorithm protocol that exhibits small estimation variance and removes estimation bias. BBC-CV’s main idea is to bootstrap the whole process of selecting the best-performing configuration based on the out-of-sample predictions of each configuration without additional training of models [32].

To implement autoML, 2D matrices were built for each disease and clinical end-points. Regarding BrCa analysis, three 2D matrices were uploaded to JADBio. A 2D matrix of 81 adjuvant patients, 17 neoadjuvant, 21 metastatic and 77 healthy controls, and 5 variables was used to classify the groups. A matrix of 119 BrCa patients and 77 healthy controls and 5 variables was applied to discriminate between BrCa and health. One last 2D matrix of 62 adjuvant patients and 19 variables was used for survival analysis. For T2D analysis, three 2D matrices were also uploaded. A matrix of 158 T2D patients, 78 prediabetic individuals and 100 healthy individuals was used to classify between groups. One matrix of 158 Diabetic patients and 100 healthy was applied to discriminate between diabetes and health. One matrix of 78 prediabetic patients and 100 healthy controls was applied to discriminate between prediabetes and health. All matrices had 5 variables. For OA analysis, one 2D matrix consisting of 17 OA patients and 17 healthy individuals and 5 variables was uploaded to the autoML platform. For all autoML analyses, we used extensive model tuning efforts. For the classification analysis, we chose the area under the curve (AUC) metric for optimization of model performance. For survival analysis, we chose the Concordance Index (CI) metric. The predictive power of each biosignature was assessed using AUC, CI and average precision (also known as area under the precision-recall curve) metrics.

## 5. Conclusions

In conclusion, in our study, ccfDNA in BrCa and Diabetes was found to be enriched by mtDNA. Also, ccf-mtDNA was correlated to clinical parameters such as hormonal receptor status on BrCa. Via an innovative machine learning tool, diagnostic and prognostic multiparameter biosignatures were produced in BrCa, Diabetes and OA. These highly performing biosignatures of diagnostic/prognostic power can add value to the clinical management of these diseases.

## 6. Patents

A patent application entitled ‘Mitochondrial fraction of circulating cell-free DNA as an indicator of human pathology’ was filed in the Hellenic Industrial Property Organization. 

## Figures and Tables

**Figure 1 ijms-25-04199-f001:**
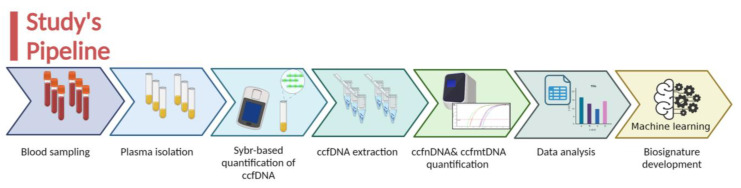
Experimental pipeline of the study.

**Figure 2 ijms-25-04199-f002:**
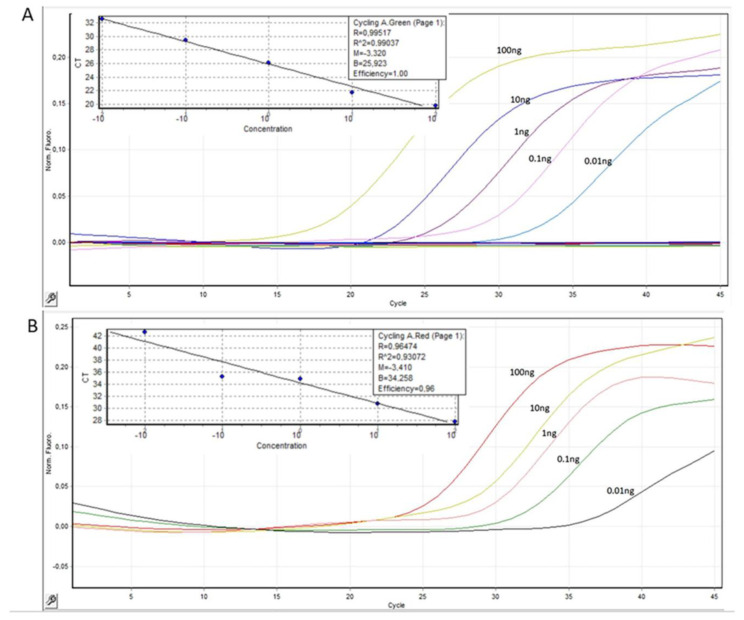
Standard and amplification curves of (**A**) mitochondrial *MTATP8* and (**B**) nuclear *GAPDH* using serial dilutions of mixed genomic DNA (100 ng, 10 ng, 1 ng, 0.1 ng, 0.01 ng).

**Figure 3 ijms-25-04199-f003:**
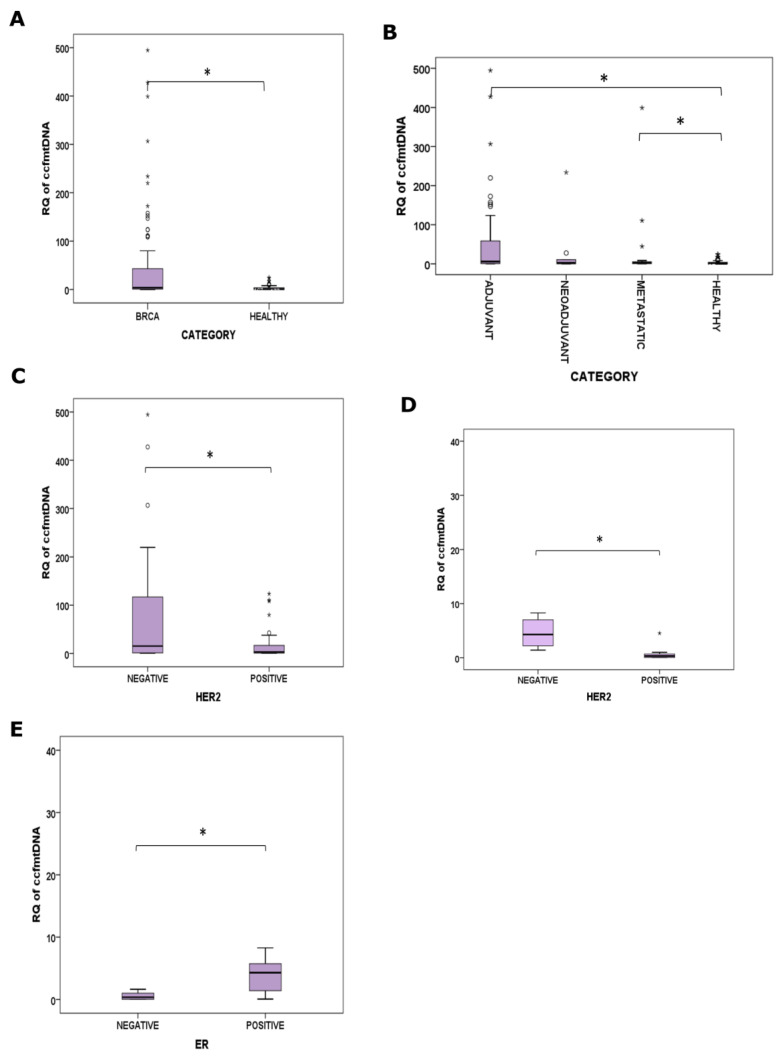
ccf-mtDNA fraction in BrCa (**A**) BrCa patients (n = 119) and Healthy individuals (n = 77), (**B**) Adjuvant (n = 81), Neoadjuvant (n = 17), Metastatic (n = 21) and Healthy groups (n = 77). (**C**) HER2 positive (n = 30) and HER2 negative (n = 51) Adjuvant group, (**D**) HER2 positive (n = 9) and HER2 negative (n = 12) Metastatic group, (**E**) ER positive (n = 13) and ER negative (n = 8) Metastatic group. Statistical significance between categories *p* < 0.05 is depicted by (*).

**Figure 4 ijms-25-04199-f004:**
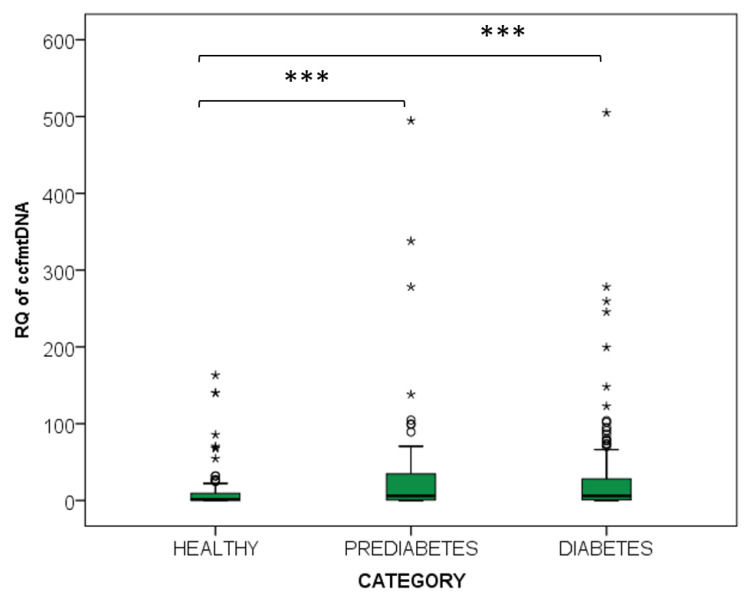
ccf-mtDNA fraction of ccfDNA in Type 2 Diabetes patients (n = 158), Prediabetics (n = 78) and Healthy individuals (100). Mild outlier values are represented by circles. Extreme outlier values are represented by asterisks. Statistical significance *p* < 0.001 in relation to the Healthy group is depicted by (***).

**Figure 5 ijms-25-04199-f005:**
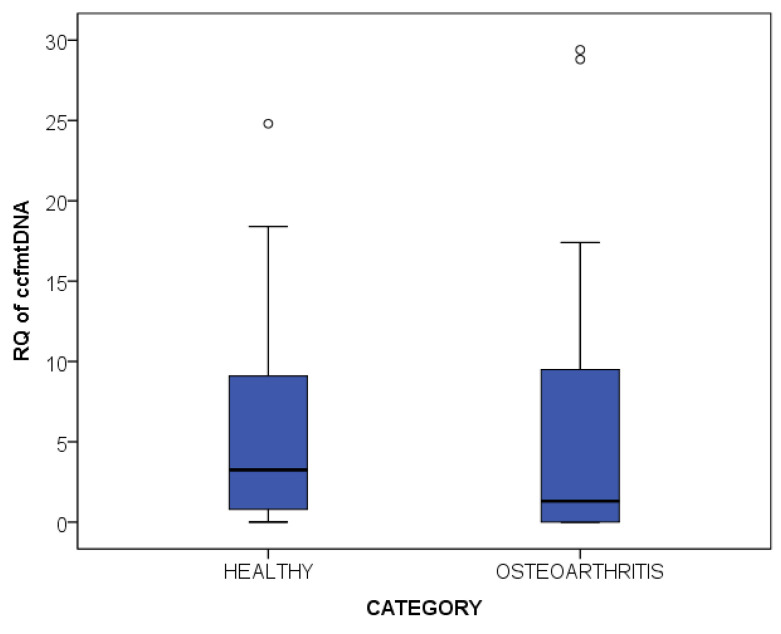
ccf-mtDNA fraction of ccfDNA in Osteoarthritis patients (n = 17) and Healthy individuals (n = 17). Mild outliers values are represented by circles.

**Figure 6 ijms-25-04199-f006:**
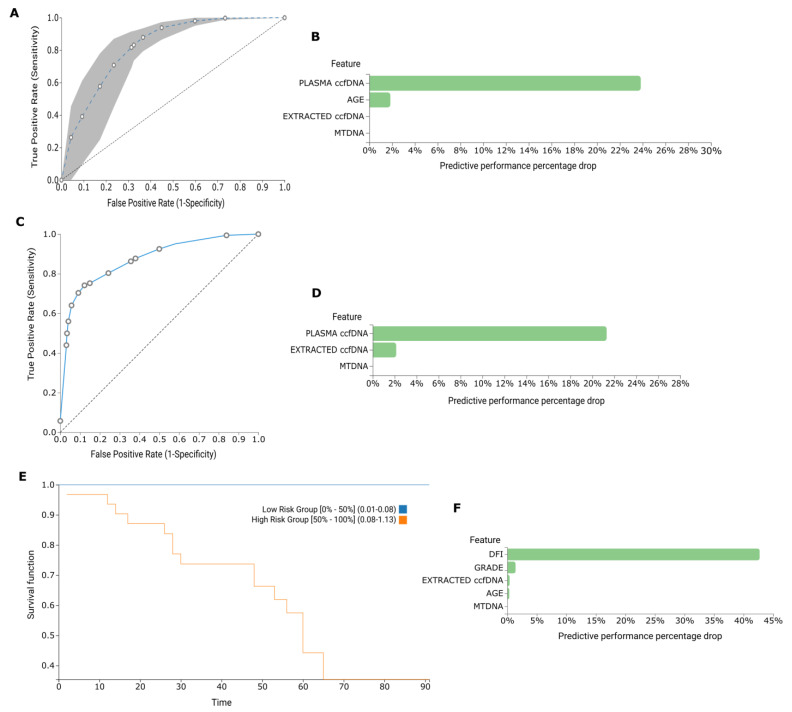
BrCa data analysis by AutoML to build diagnostic and prognostic models. ROC curve (**A**) and Feature importance plot (**B**) of built biosignature for discriminating between Adjuvant, Metastatic, Neoadjuvant and Healthy groups. ROC curve (**C**) and Feature importance plot (**D**) of biosignature for discriminating between BrCa and Health. Kaplan Mayer plot (**E**) and Feature importance plot (**F**) of a biosignature predicting overall survival in the Adjuvant group.

**Figure 7 ijms-25-04199-f007:**
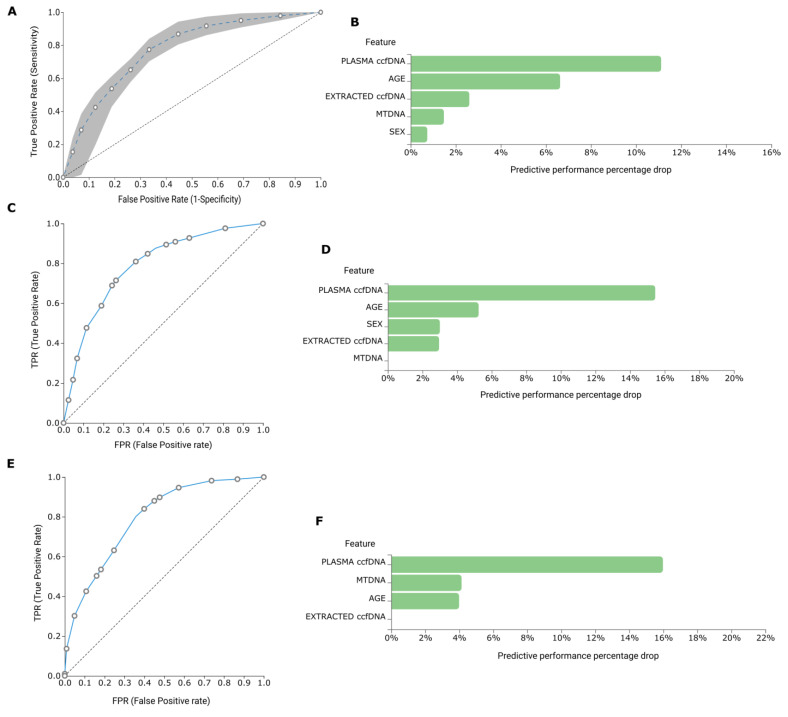
Diabetes data AutoML analysis to build diagnostic models. ROC curve (**A**) and Feature importance plot (**B**) of built biosignature for discriminating between Diabetes, Prediabetes and Healthy groups. ROC curve (**C**) and Feature importance plot (**D**) of a biosignature discriminating between Diabetes and Health. ROC curve (**E**) and Feature importance plot (**F**) of a biosignature discriminating between Prediabetes and Health.

**Table 1 ijms-25-04199-t001:** Total plasma ccfDNA and ccf-mtDNA fraction in BrCa patients and Healthy individuals.

	All BrCa (n = 119)	Adjuvant (n = 81)	Metastatic (n = 21)	Neo-Adjuvant (n = 17)	Healthy (n = 77)
Total plasma ccfDNA (ng/μL)	1035 (565–2270)	1045 (565–1800)	1030 (573–1260)	951 (667–2270)	733 (261–3720)
RQ of ccf-mtDNA	3.84 (0.01–982.00)	6.02 (0.01–982.00)	2.78 (0.01–398.00)	2.23 (0.01–962.00)	0.63 (0.01–24.77)

Abbreviations: RQ = Relative Quantity.

**Table 2 ijms-25-04199-t002:** Total plasma ccfDNA and ccf-mtDNA fraction in Diabetes, Prediabetes and respective Healthy groups.

	Diabetes Group (n = 158)	Prediabetes Group (n = 78)	Healthy Group (n = 100)
Total plasma ccfDNA (ng/μL)	989 (507–1840)	938 (439–1698)	886 (261–2800)
RQ of ccf-mtDNA	28.00 (0.01–504.90)	31.90 (0.01–494.60)	12.30 (0.01–161.10)

Abbreviations: RQ = Relative Quantity.

**Table 3 ijms-25-04199-t003:** Total plasma ccfDNA and ccf-mtDNA fraction in Osteoarthritis and Healthy groups.

	Osteoarthritis Group (n = 17)	Healthy Group (n = 17)
Total plasma ccfDNA (ng/μL)	1591 (1100–2440)	906 (319–2580)
RQ of ccf-mtDNA	6.96 (0.01–29.40)	6.34 (0.01–24.80)

**Table 4 ijms-25-04199-t004:** Demographic and clinicopathological characteristics of BrCa and respective Healthy women groups.

	Adjuvant Group (n = 81)	Metastatic Group (n = 21)	Neo-Adjuvant Group (n = 17)	Healthy Group (n = 77)
Age (years)	59 (±12)	66 (±14)	55 (±19)	56 (±13)
Menopause	44	15	6	
Stage			*	
Ι	16			
ΙΙ	32			
ΙΙΙ	26			
IV		21		
Not available	7			
Grade				
1	5			
2	34	4	8	
3	37	13	5	
Not available	5	4	4	
Lymph node status			*	
Negative	33	-	-	
Positive	40			
Not available	8			
ER status				
Positive	52	13	13	
Negative	29	8	4	
PR status				
Positive	48	13	11	
Negative	33	8	6	
Her2 status				
Positive	30	9	7	
Negative	51	12	10	

Abbreviations: ER = estrogen receptor; PR = progesterone receptor; Her2 = Human epidermal growth factor 2 receptor. * Data for lymph node status and stage are not available for the Neo-adjuvant group, as initial tumor size evaluation at the time of blood sampling was performed before surgery and chemotherapy by CT scan and pathological evaluation were performed after surgery following chemotherapy.

**Table 5 ijms-25-04199-t005:** Clinical and demographical data of Diabetes, Prediabetes and respective Healthy individuals used as control group.

	Diabetes Group (n = 158)	Prediabetes Group (n = 78)	Healthy Group (n = 100)
Age (years)	54 ± 8	51 ± 9	53 ± 10
Sex (male)	92	43	52
HbA1c (%)	6.9 (5.9–7.4)	6.1 (5.7–6.3)	5.5 (5.4–5.7)
Fasting glucose (mg/dL)	131 ± 7	117 ± 9	86 ± 11
BMI (kg/m^2^)	32.5 ± 6.2	31.8 ± 5.4	32.1 ± 6.0
Treatment	All patients receive oral or insulin medication	-	-

Abbreviations: BMI = Body Mass Index.

**Table 6 ijms-25-04199-t006:** Clinical and demographical data from Osteoarthritis and respective Healthy individuals’ groups.

	Osteoarthritis Group (n = 17)	Healthy Group (n = 17)
Age (years)	72 ± 6	67 ± 7
Sex (female)	17	17
BMI (kg/m^2^)	32.5 ± 6.2	30.2 ± 5.1
Smoking (no)	17	17
Menopause (yes)	17	17

**Table 7 ijms-25-04199-t007:** Primers and probes of dual qPCR assay.

Primer/Probe	Primer/Probe Sequence (5′–3′)
MTATP8F	ATCACCCAACTAAAAATATTAAACACAAACTA
MTATP8R	ATTTTGGTTCTCAGGGTTTGTTA
MTATP8PROBE	FAM-CTACCTCCCTCACCAAACCCATA-BHQ1
GAPDHF	TCCCCACACACATGCACTTA
GAPDHR	TAGTCCCAGGGCTTTGATT
GAPDHPROBE	CY5-GAGCTAGGAAGGACAGGCAACTT-BHQ2

## Data Availability

Data are available upon request.

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
