# Peer review of "Mitochondrial Fraction of Circulating Cell-Free DNA as an Indicator of Human Pathology"

_ijms, 2024, doi:10.3390/ijms25084199_

Round 1

Reviewer 1 Report

Comments and Suggestions for Authors

The phenomenon of circulating cell-free DNA was already described before by the same group before. This paper just listed several diseases and make them as examples for demonstration. As a research article, there is no mechanism to illustrate why circulating cell-free DNA is correlated with pathology. 

Your group already published papers with this phenomenon before, this paper did not provide too much new information. The mechanism why this circulating cell-free DNA can be related to pathology is missing, you should address this issue rather than just keep publishing the similar articles.

Author Response

The phenomenon of circulating cell-free DNA was already described before by the same group. This paper just listed several diseases and make them as examples for demonstration. As a research article, there is no mechanism to illustrate why circulating cell-free DNA is correlated with pathology.

Your group already published papers with this phenomenon before, this paper did not provide too much new information. The mechanism why this circulating cell-free DNA can be related to pathology is missing, you should address this issue rather than just keep publishing the similar articles.

Response to Reviewer 1.

The existence of cell-free DNA in the circulation is indeed known since the 40s. The reviewer correctly noted that we and others have published multiple pieces of work demonstrating aberrant levels, methylation paterns, fragment distribution, Raman spectra etc. in health and disease, in an effort to exploit this biomaterial and discover clinical utility. In this present article we focus on the mitochondrial fraction of cell-free DNA (ccf-mtDNA), i.e. this originating from mitochondria rather than the cell nucleus, as clearly stated in the title and throughout the text. We devised a novel dual qPCR assay to concurrently quantify ccf-mtDNA and ccf-nuclear DNA, offering a novel approach to its assessment. Using it, we conducted a comprehensive investigation into the potential utility of ccf-mtDNA as a biomarker across various human pathologies, including breast cancer, type 2 diabetes, and osteoarthritis, and these are original findings that have not been reported before by us or others. The reviewer seems to have overlooked the specific objective of the study and this is why its originality is questioned. We now hope that we have clarified this issue and there are no more concerns.

Reviewer 2 Report

Comments and Suggestions for Authors

This study reported mitochondrial DNA (mtDNA) releasing in three pathologies, compared the circulating cell-free DNA (ccfDNA) level between each pathology and healthy control, and uncovered a potential correlation between released mtDNA abundance and pathological progression through bioinformatic analysis. 

It is a good study that could enrich mitochondrial research in the biomedical area. The authors need to improve in manuscript preparation.  It's better to indicate "n=" in some figures. The qPCR calibration needs to show a linear curve with an error bar rather than raw data. There are also large numbers of mistyped and format issues.

Author Response

This study reported mitochondrial DNA (mtDNA) releasing in three pathologies, compared the circulating cell-free DNA (ccfDNA) level between each pathology and healthy control, and uncovered a potential correlation between released mtDNA abundance and pathological progression through bioinformatic analysis.

It is a good study that could enrich mitochondrial research in the biomedical area. The authors need to improve in manuscript preparation.  It's better to indicate "n=" in some figures. The qPCR calibration needs to show a linear curve with an error bar rather than raw data. There are also large numbers of mistyped and format issues.

We thank the reviewer for this comment and we now proofread the manuscript. Also, in the legends of figures 3, 4 and 5 we added the number of cases (n) for each group (see the following legends).

Figure 3. ccf-mtDNA fraction in BrCa A. BrCa patients (n=119) and Healthy individuals (n=77), B. Adjuvant (n=81), Neoadjuvant (n=17), Metastatic (n=21) and Healthy groups (n=77). C. HER2 positive (n=30) and HER2 negative (n=51) Adjuvant group, D. HER2 positive (n=9) and HER2 negative (n=12) Metastatic group, E. ER positive (n=13) and ER negative (n=8) Metastatic group. Statistical significance between categories p<0.05 is depicted by (*).

Figure 4. ccf-mtDNA fraction of ccfDNA in Type 2 Diabetes patients (n=158), Prediabetics (n=78) and Healthy individuals (100). Statistical significance p<0.001 in relation to the Healthy group is depicted by (***).

Figure 5. ccf-mtDNA fraction of ccfDNA in Osteoarthritis patients (n=17) and Healthy individuals (n=17).

The reviewer is correct about the missing of linear curve in figure 2. We now included the linear curve for each qPCR assay in figure 2.

Reviewer 3 Report

Comments and Suggestions for Authors

This article delineates a comprehensive investigation into the potential utility of the mitochondrial fraction of circulating cell-free DNA (ccf-mtDNA) as a biomarker across various human pathologies, including breast cancer, type 2 diabetes, and osteoarthritis. The authors devised a dual qPCR assay to concurrently quantify ccf-mtDNA and ccf-nuclear DNA, offering a novel approach to biomarker assessment. In general, the manuscript is well-crafted and easily understandable. I just have several comments that may improve the manuscript:

1. JADBio ML represents a commercial solution for data analysis. As per the website (https://jadbio.com/pricing/), the authors utilized a paid plan for the analysis conducted in the manuscript. This reliance on a proprietary tool may pose challenges for other researchers aiming to verify or replicate the findings presented in this study.

2. With the exception of Figure 5, the clarity of all other figures appears compromised, hindering comfortable readability for the audience. It is recommended to utilize clearer vector figures instead to enhance readability.

3. Regarding Figure 3, the random placement of asterisks, alternating between the Positive and Negative groups, presents confusion. It is suggested to introduce a horizontal line between the groups for comparison, thereby highlighting the statistical significance. Additionally, clarification is needed on whether one-sided or two-sided tests were employed, both in this figure and throughout the manuscript.

4. In Figure 4, it is noted that the figure name is erroneously labeled as Figure1. Furthermore, to accurately represent p-values below 0.001, (***) should be employed instead of (*).

5. Figures 6 and 7 consistently display a horizontal gray line in the feature importance plot, which appears unnecessary. It is advised to remove this element.

6. In Section 4.4, the authors provided a vague description, stating, "In case of binary variables, t-test or Mann–Whitney test were also applied. Pearson or Spearman correlation was used for comparison between two continuous variables." This ambiguity makes it challenging for readers to ascertain the specific statistical tests employed. Clarification regarding the statistical methods utilized would enhance the comprehensibility of the section.

Author Response

This article delineates a comprehensive investigation into the potential utility of the mitochondrial fraction of circulating cell-free DNA (ccf-mtDNA) as a biomarker across various human pathologies, including breast cancer, type 2 diabetes, and osteoarthritis. The authors devised a dual qPCR assay to concurrently quantify ccf-mtDNA and ccf-nuclear DNA, offering a novel approach to biomarker assessment. In general, the manuscript is well-crafted and easily understandable. I just have several comments that may improve the manuscript:

  1. JADBio ML represents a commercial solution for data analysis. As per the website (https://jadbio.com/pricing/), the authors utilized a paid plan for the analysis conducted in the manuscript. This reliance on a proprietary tool may pose challenges for other researchers aiming to verify or replicate the findings presented in this study.

Indeed, JADBio offers a commercial solution but its basic edition offers a free full functional tool for data analysis. For each analysis presented in the article a link is provided and a reader or researcher can verify results, as it gives access to a full report to the pipeline and the results produced, such as all the algorithms applied and preprocessing conducted, the performance overview, the feature selection and existing alternatives and all analysis visualization such as UMAP plot or PCA plot. The link works by pasting the link into a browser.  

  1. With the exception of Figure 5, the clarity of all other figures appears compromised, hindering comfortable readability for the audience. It is recommended to utilize clearer vector figures instead to enhance readability.

We thank the reviewer for the comment and we apologize for the inconvenience. Now, the quality of figures was significantly improved.

  1. Regarding Figure 3, the random placement of asterisks, alternating between the Positive and Negative groups, presents confusion. It is suggested to introduce a horizontal line between the groups for comparison, thereby highlighting the statistical significance. Additionally, clarification is needed on whether one-sided or two-sided tests were employed, both in this figure and throughout the manuscript.

We have now added horizontal lines and asterisks where necessary in Figure 3.

Also, in the Methods section specifically in line 447, we have added the following “All statistical tests employed in our analysis were two-sided.” in order to clarify the type of statistical tests.

  1. In Figure 4, it is noted that the figure name is erroneously labeled as Figure1. Furthermore, to accurately represent p-values below 0.001, (***) should be employed instead of (*).

We apologize for the inconvenience. Now, p-values below 0.001 were corrected to (***) as well as the figure’s name.

  1. Figures 6 and 7 consistently display a horizontal gray line in the feature importance plot, which appears unnecessary. It is advised to remove this element.

 We thank the reviewer for the comment. The horizontal gray lines were removed from figures 6 and 7.

  1. In Section 4.4, the authors provided a vague description, stating, "In case of binary variables, t-test or Mann–Whitney test were also applied. Pearson or Spearman correlation was used for comparison between two continuous variables." This ambiguity makes it challenging for readers to ascertain the specific statistical tests employed. Clarification regarding the statistical methods utilized would enhance the comprehensibility of the section.

This is an important concern. We now clarify each specific test used in every single analysis in the results section as shown in the following lines.

Line 92: ‘BrCa patient groups as compared to health (p=0.005, Mann-Whitney U=564). Interestingly, those increased ccfDNA levels in BrCa were also more abundant in their ccf-mtDNA content, as the later was found to be higher in the BrCa patients as compared to healthy individuals (p<0.001, Mann-Whitney U=2181) (Table 5, Figure 3A). In particular, in the Adjuvant and Metastatic BrCa groups, ccf-mtDNA content was statistically significantly higher (p<0.001, Mann-Whitney U=1282 and p=0.044, Mann-Whitney U=429, respectively) (Figure 3B) as compared to health, the group…’

Line 103: ‘…absence of HER-2/neu expression was correlated with higher ccf-mtDNA fraction (p=0.05, Mann-Whitney U=364) (Figure 3C). Similarly, in the Metastatic group, elevated ccf-mtDNA content was correlated to the absence of HER-2/neu expression (p=0.006, Mann-Whitney U=5) (Figure 3D) and to the presence of ER receptor (p=0.028, Mann-Whitney U=7) (Figure 3E).’

Line 114: ‘… both patient groups as compared to the healthy individuals (p<0.001, Kruskal-Wallis df=2 )…

Line 117: ‘…relation to Healthy individuals (p=0.008, Mann-Whitney U=1591 and p<0.001, Mann-Whitney U=821, respectively)…’

Line 129: ‘… to the Healthy group (p<0.001, t-test df=29). Ccf-mtDNA content didn’t differ between OA and Health (p=0.852, t-test df=29) (Figure 5).’

Also, in the Methods section we indicate what type of correlation was used in our analysis using either R for Pearson correlation or r for Spearman correlation as follows:

Line 448: ‘Pearson (R) or Spearman (r) correlation was used for comparison between two continuous variables.’

Round 2

Reviewer 1 Report

Comments and Suggestions for Authors

As I mentioned, this phenomenon (mitochondria DNA derived cell free DNA) was found previously. This article simply demonstrates a method to measure and "correlate" mt-cfDNA to diseases rather than a mechanism to describe it. This article should be published on "methodology" journal rather than regular articles.   

Reviewer 3 Report

Comments and Suggestions for Authors

The authors have made their best efforts to address my comments from the previous round of review. I have no more comments.